# Adsorption of Toxic Tetracycline, Thiamphenicol and Sulfamethoxazole by a Granular Activated Carbon (GAC) under Different Conditions

**DOI:** 10.3390/molecules27227980

**Published:** 2022-11-17

**Authors:** Risheng Li, Wen Sun, Longfei Xia, Zia U, Xubo Sun, Zhao Wang, Yujie Wang, Xu Deng

**Affiliations:** 1Shaanxi Provincial Land Engineering Construction Group Co., Ltd., Xi’an 710075, China; 2Key Laboratory of Degraded and Unused Land Consolidation Engineering, The Ministry of Natural Resources, Xi’an 710075, China; 3Department of Environmental Science and Engineering, School of Energy and Power Engineering, Xi′an Jiaotong University, Xi’an 710049, China; 4Department of Chemistry, College of Resource and Environment, Baoshan University, Baoshan 678000, China; 5School of Basic Medicine, Shaanxi University of Chinese Medicine, XiXian New Area, Xianyang 712046, China

**Keywords:** adsorption, antibiotics, activated carbon, water treatment

## Abstract

Activated carbon can be applied to the treatment of wastewater loading with different types of pollutants. In this paper, a kind of activated carbon in granular form (GAC) was utilized to eliminate antibiotics from an aqueous solution, in which Tetracycline (TC), Thiamphenicol (THI), and Sulfamethoxazole (SMZ) were selected as the testing pollutants. The specific surface area, total pore volume, and micropore volume of GAC were 1059.011 m^2^/g, 0.625 cm^3^/g, and 0.488 cm^3^/g, respectively. The sorption capacity of GAC towards TC, THI, and SMZ was evaluated based on the adsorption kinetics and isotherm. It was found that the pseudo-second-order kinetic model described the sorption of TC, THI, and SMZ on GAC better than the pseudo-first-order kinetic model. According to the Langmuir isotherm model, the maximum adsorption capacity of GAC towards TC, THI, and SMZ was calculated to be 17.02, 30.40, and 26.77 mg/g, respectively. Thermodynamic parameters of Δ*G*^0^, Δ*S*^0^, and Δ*H*^0^ were obtained, indicating that all the sorptions were spontaneous and exothermic in nature. These results provided a knowledge base on using activated carbon to remove TC, THI, and SMZ from water.

## 1. Introduction

Antibiotics can be used to kill or inhibit the growth of bacteria. Although antibiotics are used in humans and animals, roughly 80% of their total usage is on livestock and poultry for human consumption. Antibiotics are routinely added to the food and water of livestock to promote growth and improve feed-use efficiency. In addition, antibiotics are injected into animals when they are sick or at high risk of getting sick. According to recent sales data of the world market, China is the biggest producer and user of antibiotics [1]. When antibiotics are used, the organisms cannot absorb antibiotics fully, so they are released into the environment in an active form [2]. In 2013 in China, more than 50 × 10^3^ tons of antibiotics entered into aquifers, as stated in a report. The aquifers include the outflow of sewage processing plants [3], drinking water, groundwater [4], rivers and lakes [5], and seawater. Antibiotics have different half-lives, which some are long-lived [6], and their contagion rates in the environment have increased over the years. In a water-based environment, antibiotics are generally harmful, i.e., they prevent the ability to break down micro-organisms deposits, destructs the development of marine organisms as well encourage maturation in bacterial drug-opposing genetic codes [7]. Various research [8] have shown that any contact with antibiotics (μg/L–mg/L) creates an adverse effect and influence on the lives of water-based creatures, for example, the growth of their body and weight. 

The residual of these antibiotics in water-based items comes into the human body with the help of the food chain and later mounts up via biological enhancement [9]. Since the majority of antibiotics are cancerogenic, teratogenic, and mutagenic, along with creating hormone-related issues, so using antibiotics causes serious interference with the anatomy of humans and the immune system [7]. Antibiotics are obtaining the identification of rising environmental contaminants, are classified as fractious bio-accumulative substances [10,11], as well as are considered harmful and toxic chemicals.

The outflow of Municipal treatment plants and pharmaceutical manufacturing plants are the basic sources of discharging antibiotics in water. According to Michael [12] and Rizzo [13], the cities’ wastewater treatment plants are considered the main source of releasing antibiotics into the environment. It is important to dispose of residues of antibiotics before discharging wastewater into the environment. There is an urgent need for case studies to provide a cheap solution for eliminating antibiotics. Sera Budi Verinda et al. used ozonation to remove ciprofloxacin in wastewater, and the removal rate reached 83.5% [14]; Shang, K.F. et al. indicate that the combination of DBD plasma and PMS/PDS is an efficient pretreatment method for bio-treatment of refractory SMX [15].

The adsorption process is [16] easy to plan and feasible to function. The adsorbent should be a biomass adsorbent, such as agricultural waste, which is environmentally friendly and economical [17]. The adsorption method is utilized to remove organic pollutants from contaminated waters over the surface of the adsorbent [18]. Its application to eliminating antibiotics for approximately 30 different compounds has been reported so far [19]. The performance of adsorption processes is largely influenced by hydrogen bonding and electrostatic interactions. Different adsorbent materials are used for the elimination of antibiotics from the aqueous solution, for instance, clinoptilolite [20], soil [21], different kinds of activated carbons [22,23,24,25], calcium phosphate materials, and core-shell magnetic nanoparticles [26]. Compared with traditional adsorption materials, activated carbon has excellent porosity, large specific surface area, low cost, and environmental friendliness and is reported to be an effective adsorbent for eliminating trace pollutants [27]. Granular activated carbon is divided into stereotyped and unshaped particles. It is mainly made of coconut shell, nut shell, and coal, which is refined through a series of production processes. Its appearance is black amorphous particles; it has developed pore structure, good adsorption performance, high mechanical strength, low cost, etc. Therefore, granular activated carbon is widely used in drinking water, industrial water, wine, waste gas treatment, decolorization, desiccant, gas purification, and other fields [28,29].

At present, there is very little related research on antibiotics in wastewater through activated carbon, In this paper, three different kinds of antibiotics, including tetracycline, thiamphenicol, and sulfamethoxazole, were selected as the target pollutants, and a kind of activated carbon in granular form was used as the adsorbent. The adsorption capacity and mechanism were studied. This paper aimed to evaluate the removal efficiency of adsorption technology in treating antibiotics-loading wastewater and to promote the application of activated carbon in such a field.

## 2. Results and Discussion

### 2.1. Characterization of the Granular Activated Carbon

Figure 1A exhibits the outlook of GAC with an average length of 1–2 mm and diameter of 1 mm. Nitrogen adsorption/desorption at 77 k for the granular activated carbon was shown in Figure 1B, in which the GAC sample possessed a type I sorption isotherm. The details of BET are shown in Table 1: These results demonstrated that GAC was of high specific surface area and porous structure. In conclusion, this type of GAC might be an ideal adsorbent for removing antibiotics from wastewater. In Figure 1A, the SEM image of granular activated carbon shows that the external surface of GAC was multi-layer and rigid, which helped to increase the specific surface area of granular activated carbon. The measurement of zeta potential is a technique for calculating the surface charge of activated carbon in a colloidal solution. The graph of activated carbon zeta potential in the solution was shown as a function of pH in Figure 1B. The graph shows that the surface charge of activated carbon was linked with solution pH. The pH_ZPC_ zero-point charge of activated carbon is 4.0, indicating the charge on the activated carbon surface was positive when the solution pH was less than pH_ZPC_ while negative when the solution pH was greater than pH_ZPC_. It can be seen from Figure 1D that the infrared spectrum of GAC has a C-O characteristic absorption peak near 1000 cm^−1^, a C=C characteristic absorption peak near 1600 cm^−1^, and an O-H stretching vibration peak near 3200 cm^−1^.

### 2.2. Effect of pH on GAC Adsorption

Under the conditions of environmental conditions of 25 °C, the antibiotic concentration of 25 mg/L, GAC dosage of 8 g/L, TC adsorption time of 100 min, and THI and SMZ adsorption time of 60 min: when the pH of the solution is 7, the effect of GAC on TC, THI, and SMZ the best adsorption efficiencies are 91.76%, 96.34%, and 94.23%, respectively (Figure 2).

### 2.3. Effect of GAC Dosage on the Adsorption

Figure 3 shows the removal efficiency versus GAC dosage. When the GAC dosage increased from 2 to 8 mg/g, all the removal efficiencies for the three antibiotics increased remarkably and then slightly from 6 to 10 mg/g. A possible reason was that increased GAC dosage increased the sorbent surface area, the number of sorption sites, and the contact area increased [30,31,32]. Considering the removal efficiencies and economic benefits, this study selected a GAC dosage of 4 mg/g as an optimum dosage.

### 2.4. Determination of the Adsorption Equilibrium Time

For the measurement of equilibrium time, *q_t_* versus contact time (*t*) was represented in Figure 3. During the adsorption of TC, it was seen that the value of *q_t_* increased quickly within the first 30 min and then slightly from 30 to 100 min for all the initial concentrations.

The possible cause was that there were enough adsorption sites on the surface of GAC during the initial stage of adsorption (0–30 min). As the contact time prolonged, the adsorption sites provided by GAC became fewer and fewer. As a result, *q_t_* increased slightly (30–100 min). In the case of SMZ, the value of *q_t_* increased significantly in the first 20 min, and from 20 to 60 min, the *q_t_* value increased slightly. Different from TC and SMT, the adsorption process of THI was faster, in which after 10 min, no increase in the value of *q_t_* was observed. Therefore, the equilibrium times for TC, SMT, and THI were set at 100 min, 60 min, and 10 min. In most cases, when the initial concentration of antibiotic was low while the removal efficiency was high (Figure 4), the possible reason was that at low concentrations, there are more available adsorbing sites for antibiotic molecules to absorb onto.

### 2.5. Sorption Kinetics

The pseudo-first-order and pseudo-second-order models were used to describe all the data shown in Figure 3. The results are given in Table 2, in which Equations (1) and (2) represent the mathematical formula of the two models.
(1)$$\ln(q_{e,\exp}-q_{t})=\ln(q_{e,cal})-k_{1}t$$
(2)$$\frac{t}{q_{t}}=\frac{1}{k_{2}q_{e,cal}^{2}}+\frac{1}{q_{e,cal}}t$$
where

*q_e_*,_exp_ = adsorption amount in (mg/g) at equilibrium

*q_t_* = adsorption amount in (mg/g) at time *t*

*k*_1_ = rate constant of pseudo-first-order (min^−1^)

*k*_2_ = rate constant of pseudo-second-order [g/(mg min)].

**Table 2 molecules-27-07980-t002:** Parameters of the pseudo-first-order and pseudo-second-order kinetic models for the sorption of the three antibiotics onto GAC at 25 °C.

			Pseudo-First-Order Model			Pseudo-Second-Order Model			
Sorbate	*C*_0_ (mg/L)	*q_e,exp_* (mg/g)	*q_e,cal_* (mg/g)	*K*_1_ (min^−1^)	*R* ^2^	*q_e,cal_* (mg/g)	*K*_2_ [g/(min mg)]	*K*_2_*q*^2^*_e,cal_* [mg/(min g)]	*R* ^2^
TC	6.01	0.52	2.332	0.023	0.77	0.49	0.704	0.169	0.68
	12.02	1.27	0.481	0.043	0.88	1.64	0.028	0.076	0.91
	24.04	2.78	2.773	0.053	0.98	3.46	0.014	0.167	0.98
	48.09	5.75	3.377	0.050	0.94	6.27	0.020	0.790	0.99
THI	7.12	0.79	15.7	0.109	0.73	0.80	5.14	3.306	1.00
	10.68	1.24	14.11	0.100	0.74	1.25	2.39	3.740	1.00
	17.81	2.13	12.23	0.091	0.76	2.14	2.65	12.158	1.00
	35.62	4.35	10.51	0.096	0.75	4.37	1.54	29.409	1.00
SMZ	6.33	0.63	4.57	0.021	0.78	0.65	0.811	0.349	0.99
	12.66	1.42	3.38	0.027	0.77	1.46	0.482	1.033	0.99
	25.32	3	1.11	0.038	0.84	3.10	0.185	1.781	0.99
	50.65	6.16	1.58	0.074	0.95	6.4	0.78	31.948	0.99

Figure 4 shows the linearized graph of *t*/*q_t_* versus the time of pseudo-second-order kinetic. Table 2 contains all the parameters of these two kinetic models. The value of *R*^2^ obtained from the pseudo-second-order model is given in Table 2, which is close to unity, indicating that the pseudo-second-order kinetic model best fits the adsorption of antibiotics on GAC. Furthermore, the experimental value of *q_e_*,_exp_ in (mg/g) agreed with the calculated value of *q_e_*_,*cal*_ (mg/g). These results indicate that the rate of adsorption on GAC is controlled by chemisorption. At the same time, valency forces with an exchange, or there might be a sharing of the electrons in-between these four antibiotics and the GAC. According to Table 2 primary initial adsorption rate *K*^2^*q*^2^*_e,cal_* gradually increased with the concentration of the primary four antibiotics, indicating that the greater value of concentration enhanced driving forces that can help to overcome the barrier of mass transfer-resistance in between the phases of solid and liquid.

Weber and Morris’s intraparticle diffusion model was utilized to analyze the experimental data to determine the rate-limiting step during the adsorption process. All values are shown in Table 2.
(3)$$q_{t}=k_{id}t^{1/2}+I$$
where

*q_t_* (mg/g) = the removal amount at time *t* and reaction equilibrium

*k_id_* (mg/g min^1/2^) = the particle diffusion rate constant

*I* (mg/g) = intercept, which gives the information about the boundary layer effect. If the value of *I* is greater than the boundary layer has a greater effect.

If the plot between *q_t_* versus *t*^1/2^ is linear, then intraparticle diffusion takes place. When the plot also passes through the origin (*I* = 0), the rate-limiting is controlled by intraparticle diffusion. If the plots showed deviation from linearity, then it indicates the effect of the boundary layer. Figure 5 represents the plot of intraparticle for three antibiotics. This was observed that during whole time plots, the linear portion could not pass through the origin, indicating that both boundary layer and intraparticle diffusion occur during the adsorption of antibiotics on GAC.

In Table 3, the rate constant and intercept values are shown. The intercepts C of the straight lines fitted by TC, THI, and SMZ are not 0, indicating that internal diffusion is not the only step controlling the removal of antibiotics by GAC, and the adsorption rate should be controlled by both external diffusion and intraparticle diffusion. According to Figure 5, the plots of TC and SMZ have two different portions. Firstly, the steeper segment of the plot depicts external surface adsorption, and the second portion, slow adsorption, shows intraparticle diffusion.

Boyd’s kinetic model was used to further examine the kinetic data to measure the slowest step involved in the adsorption process.
(4)$$F(t)=1-(\frac{6}{\pi })\sum_{n=1}^{\infty }\left(\frac{1}{n^{2}}\right)\exp(-n^{2}B_{t})$$
where

*F*(*t*) *= q_t_*/*q_e_* = ratio of the antibiotics adsorbed at time *t* and equilibrium;

*B_t_* = function of *F*(*t*)

If the *F*(*t*) value is higher than 0.85, then
(5)$$B_{t}=0.4977-\ln(1-F(t))$$

If the *F*(*t*) value is less than 0.85, then
(6)$$B_{t}={\left(\sqrt{\pi }-\sqrt{\pi -\left(\frac{\pi^{2}F(t)}{3}\right)}\right)}^{2}$$

As for the Boyd kinetic model, if the plot of *B_t_* against *t* is linear and through the origin, it suggests that intraparticle diffusion controls the process of mass transfer. The sorption rate is controlled through film diffusion when the plot can be seen as nonlinear or linear but does not go through the origin. Figure 6 illustrates the Boyd plots for the three antibiotics on the GAC sample. The fact that Boyd plots were linear even though they were unable to pass through the origin was noticed. These results indicate the information regarding the fact that the major controlling process required for the adsorption procedure was, nonetheless, diffusion with the layer at the border. Table 4 lists the parameters of the Boyd kinetic model.

### 2.6. Sorption Isotherm

The adsorption process can be defined as the process of mass transfer of adsorbate at the boundary layer in-between the solid adsorbent and liquid phase. The adsorption isotherm is defined as the equilibrium relationship between solid adsorbent and adsorbate at a constant temperature. For example, the ratio between the amount of adsorbate absorbed on the solid and the remaining amount left in the aqueous solution at equilibrium. There are several adsorption isotherms models like Langmuir, Freundlich, and Temkin.

Experimental data can be fitted using these models to examine the suitability of the model. The information obtained from these models can be used for designing the adsorption process. The parameters of adsorption isotherm normally estimate the sorption ability of several adsorbents for specific adsorbates with predetermined reaction conditions. The performance of the sorption process depends not only on the rate at which mass transfer occurs but also on the sorbent–sorbate equilibrium concentration [33]. Three different isotherm models, called the Langmuir [34], Freundlich [35], and Temkin [21,22], were used. The mathematical formula of these isotherm models was shown as follows:(7)$$q_{e}=\frac{q_{m}k_{L}C_{e}}{1+k_{L}C_{e}}$$
(8)$$q_{e}=k_{F}C_{e}^{1/n}$$
(9)$$q_{e}=B\ln(k_{T}C_{e})$$

*k_L_* = constant from Langmuir (L/mg)

*C_e_* = adsorbate residual concentration (mg/L)

*q_e_* = The amount of adsorbate per unit mass of sorbent (mg/g)

*q_m_* = The maximum sorption capacity (mg/g)

*K_F_* (mg/L^(1−1/n)^ g) and n are constants of the Freundlich model

*B* is the Temkin constant of the sorption heat, and *K_T_* (1/mg) stands for the constant of the Temkin isotherm.

Both Table 5 and Figure 7 show that the Langmuir model is best fitted to the experimental data than Freundlich and Temkin. The calculated values of *q*_max_ from Langmuir indicate that the maximum degree of adsorption capacity of the GAC sample to three antibiotics has been following the trend: THI > SMZ > TC.

In 1974, Weber and Chakkravorti explained the Langmuir isotherm. The formula is given below.
(10)$$R_{L}=\frac{1}{1+k_{L}C_{0}}$$
where,

*k_L_* = constant from Langmuir model (L/mg)

*C*_0_ = adsorbate initial concentration (mg/L)

*R_L_* is a separation factor that explains the nature of adsorption, and an explanation of this is given in Table 5 and Table 6. According to this formula, the value of *R_L_* in all this experiment is 0 < *R_L_* < 1, which suggests the favorable nature of the adsorption of antibiotics.

### 2.7. Adsorption Thermodynamics

The Gibbs free energy change (Δ*G*^0^) is an indication of the spontaneity of a chemical reaction and therefore is one of the most important criteria. It is calculated as follows:(11)$$\Delta G^{0}=-RT\ln K$$
where *R* is the universal gas constant (8.314 J/(molK)), *T* is the absolute temperature in (K), and *K* is the thermodynamic equilibrium constant [36].

And
(12)$$\Delta G^{0}=\Delta H^{0}-T\Delta S^{0}$$

After combining Equations (13) and (14), we get
(13)$$\mathrm{Ln}K_{L}=\frac{-\Delta H^{0}}{R}\times \frac{1}{T}+\frac{\Delta S^{0}}{R}$$

By constructing a plot of Ln*K_L_* versus 1/*T*, from the intercept calculated the change in entropy (Δ*S*^0^) and by the slope, it is possible to calculate the change in enthalpy (Δ*H*^0^) [37].

The value of *K_L_* is obtained from the Langmuir isotherm model. The value of *K_L_* is in (L/mg) so first, convert it into (L/g) by multiplying 1000 and then multiply it by antibiotic molecular mass. The value of *K_L_* then becomes dimensionless because, for measuring the correct value of thermodynamics parameters, we need *k* in the dimensionless unit [38,39].

The thermodynamics parameters Δ*G*^0^, Δ*H*^0^, and Δ*S*^0^ are shown in Table 7, and also plot between ln*K* versus 1/*T* is shown in Figure 8. The Δ*G*^0^ negative values verified the process feasibility and the spontaneous nature of adsorption. As a rule of thumb, the decrease in the negative value of Δ*G*^0^ with temperature increase indicates that the adsorption at higher temperatures is more favorable. This may be possible because, with the increase in temperature, the mobility of the adsorbate ion/molecule in the solution increases, and the adsorbate affinity to the adsorbent is high. On the contrary, an increase in the negative value of Δ*G*^0^ with an increase in temperature implies that lower temperature facilitates adsorption.

The negative value of enthalpy Δ*H* suggests that the sorption of three antibiotics is exothermic in nature. The positive value of Δ*S*^0^ indicates high randomness at the solid/liquid phase with some structural changes in the adsorbate and the adsorbent. The negative value of Δ*S*^0^ suggests that the adsorption process is enthalpy driven. The negative value of entropy change (Δ*S*^0^) also means that the disorder of the solid/liquid interface decreases during the adsorption process, resulting in the escape of adsorbed ions/molecules from the solid. Therefore, the amount of adsorbate adsorbed will decrease.

*K* is the adsorption equilibrium constant of Langmuir isotherms. The value of *k* is in the unit (L/mg). First, it converts it into (L/g) and then multiplies it by the molecular formula of the antibiotic. Its value then becomes dimensionless, as we required a dimensionless value of *k* for measuring the value of Δ*G* (Table 1).

### 2.8. Regeneration Experiments

Through regeneration experiments, it was found that the adsorption capacity of granular activated carbon for TC, THI, and SMZ decreased with the increase in regeneration times (Figure 9). At an ambient temperature of 25 °C, the initial concentration of antibiotics was 25 mg/L, and the dosage of GAC was 8 g/L. The adsorption of TC on GAC reached saturation after 60 min. The maximum adsorption efficiencies of initial GAC, 1 GAC regeneration, and 5 regeneration GAC were 92.54%, 85.73%, and 62.14%. Under the same conditions, the maximum adsorption efficiencies of initial GAC, 1-time regeneration GAC, and 5-time regeneration GAC for THI and SMZ were: 96.32%, 32.33%; 91.23%, 85.21%; 70.26%, 50.19%, respectively. Compared with other adsorbents, such as attapulgite, granular activated carbon has the characteristics of easy preparation and better regeneration [40].

## 3. Materials and Methods

### 3.1. GAC Preparation

The Granular activated carbon used in these experiments was prepared using corn Stover collected from Shaanxi agriculture Technology Company (Xi’an, China). First, this corn Stover was put 1 month for air drying. After, it was chopped into small pieces of length nearly 5 cm and positioned inside an electrically operating container resistance furnace (LNB4-13Y; Haozhuang Co., Ltd., Shanghai, China). Under a nitrogen atmosphere, pieces of cornstalk were heated at 500 °C for 2 h and then 700 °C for 2 h.

Physical activation management was carried out in an activation furnace (HHL-1; Huatong Co., Ltd., Zhengzhou, Henan, China) through superheated steam at elevated temperatures (600 °C, 2.0 MPa) for 2 h to obtain the AC. The AC was also cleansed using 1.0 M HCl-HF (1:1, *v*/*v*) solution thrice and with ultrapure water a few times till the 7.0 pH value was obtained. After washing, AC drying was carried out at a temperature of 100 °C for a duration of 15 h and then crushed and pushed through a nylon mesh with an entrance of 5.0 mm. Finally, the AC’s morphology was granular.

### 3.2. Chemicals

The physio-chemical characteristics and molecular structures of tetracycline (TC), thiamphenicol (THI), and sulfamethoxazole (SMZ) are given in Table 8. All the antibiotics have chromatographic clarity, purchased from J & K Scientific (Beijing, China). *N*,*N*-dimethylformamide with more than 99.9% purity was purchased by Sigma-Aldrich (Shanghai, China). Apparatus SPI-11-10T was used to prepare ultra-pure water (ULUPURE, Chengdu, Sichuan, China). The GAC sample used in this paper was supported by Fan et al. [41].

### 3.3. Methodology for Selecting Antibiotics

Wastewater influents contain several types of antibiotics; however, due to the limited availability of information, only a few antibiotics will be selected for this work. The criteria for selecting antibiotic classes were defined by considering (1) the relevance of antibiotic class to human medicine, (2) usage amongst the different animal species, and (3) their presence in wastewater treatment plants.

Based on the selection criteria mentioned above, the following antibiotics were selected for this work: (1) Tetracycline, (2) Thiamphenicol, and (3) Sulfamethoxazole.

### 3.4. Sorption Experiments

**Stock solution preparation:** 0.1000 g of TC, THI, and SMZ was dissolved in a 50 mL volumetric flask with ultra-pure water and then transferred into a 100 mL volumetric flask to obtain the stock solutions with a final concentration of 1 g/L respectively. The testing solutions with different concentrations were obtained by diluting the stock solutions with ultrapure water.

**Effect of GAC dosage:** on the sorption of TC by the GAC, tests were conducted in a 100 mL beaker containing 50 mL TC testing solution with 25 mg/L. For GAC dosage, the amount of GAC ranged from 2 to 8 g/L, while the temperature was fixed at 25 °C and the contact time was 100 min. The effect of GAC dosage was also conducted for the sorption of THI and SMZ.

**Determination of the adsorption equilibrium time:** to investigate the sorption of TC onto the GAC, batch experiments were carried out using a 100 mL beaker containing 0.4000 g GAC and 50 mL TC testing solutions with various preliminary concentrations. Each beaker was put into a thermostatic reciprocating shaker (ZHWY-2102C, Changzhou Guowang Instrument Manufacturing Co., Ltd., Changzhou, Jiangsu, China) at 180 r/min and 25 °C in the dark. The sample was withdrawn by a 5 mL syringe from one beaker after shaking for 5 min and passed through a 0.45 µm filter. The sampling times were 5, 10, 15, 20, 30, 40, 50, 60, 80, and 100 min. The same steps were carried out to investigate the sorption of SMZ and THI on GAC, respectively. Filter liquor was used to determine the residual concentrations of the three antibiotics.

**Adsorption isotherm:** for measuring the adsorption isotherm, a 250 mL beaker containing 50 mL testing solution with a different initial concentration of TC and 0.4000 g of GAC was agitated in the dark under 180 rpm at 25 °C. Samples were withdrawn after 100 min using a 5 mL pipette tip and filtered using a 0.45 μm filter membrane. This filter-out solution of TC was used for the remaining concentration analysis to find adsorption isotherm representation. For investigating the adsorption isotherm of THI and SMZ, the same procedure was followed.

**Adsorption Thermodynamics:** a 250 mL beaker containing 50 mL testing solution with 25 mg/L of TC and 0.4000 g of GAC was agitated in the dark under 180 rpm at 15, 20, and 25 °C. Samples were withdrawn after 100 min using a 5 mL pipette tip and filtered using a 0.45 μm filter membrane. This filter-out solution of TC was used for the remaining concentration analysis. For investigating the adsorption isotherm of THI and SMZ, the same procedure was followed.

**Analysis:** physisorption analyzer (ASAP-2020, Micrometrics, Beijing, China) was used to measure the prepared GAC characteristics, like surface area, pore volume, and pore diameter at 77 K temperature. The Brunauer-Emmett-Teller (BET) method was used to measure surface area, and the Barrett-Joyner-Halenda (BJH) method was used to measure pore size distribution. Scanning electron (TM-1000; Hitachi, Tokyo, Japan) was used to describe the GAC surface morphology and porous structure. The Zeta potential analyzer (Malvern Zetasizer Nano S90, Shanghai, China) was used to monitor GAC Zeta potential value. The GAC was first converted into powder form and then submerged in NaCl solution (1 mmol = litter) to make a blend (0.1 g = 1 L). To adjust the solution’s pH value from 3 to 9, HCl or NaOH was used. This solution was placed in an ultrasonic treatment apparatus (25 °C, 40 kHz) for 30 min. After the ultrasonic treatment, this solution was kept for 24 h, and its zeta potential value was measured using a zeta potential analyzer. The Ultraviolet-visible spectrophotometer (model: SP-1915, Spectrum, Shanghai, China) was used to measure the antibiotics’ residual concentration. The calibration curves for TC, SMZ, and THI were given as y = 0.0324x + 0.0307, y = 0.0635x + 0.0007, and y = 0.1455x + 0.0068, respectively. For measuring the sorption capacity on GAC following formula can be used.
(14)$$q_{t}=\frac{C_{0}-C}{M}\times V$$
where

*q_t_* = adsorption capacity at time *t* (mg/g)

*C*_0_ = adsorbate initial concentration in (mg/L)

*C* = adsorbate residual concentration at time *t* (mg/L)

*V* = volume of solution (L)

*M* = mass of sorbent (g)

The removal efficiency at different initial concentrations of antibiotics was calculated by using the following formula:(15)$$R.E=\frac{C_{0}-C_{E}}{C_{0}}\times 100\%$$
where

*R.E* = removal efficiency (%)

*C*_0_ = initial concentration (mg/L)

*C_E_* = equilibrium concentration (mg/L)

### 3.5. Regeneration Experiments

Saturated GAC (8.0000 ± 0.0004) g adsorbing TC, THI, and SMZ was placed in a quartz glass reactor, and N_2_ was used as a protective gas and placed in a microwave oven for irradiation, microwave power 730 w, microwave time 180 s [42], and carried out microwave regeneration test.

## 4. Conclusions

This research investigated the experimental results of granular activated carbon adsorbing different antibiotics, including tetracycline, thiamphenicol, and sulfamethoxazole, from an aqueous solution. The BET experiment indicated that GAC had a high specific surface area of approximately 1059 m^2^/g and a high pore volume of 0.625 cm^3^/g, meaning that it is a very useful adsorbent for antibiotics removal. The equilibrium adsorption data of TC, THI, and SMZ were well expressed by the Langmuir isotherm model, and maximum adsorption capacities were 17.02, 30.40, and 26.77 mg/g, respectively. The kinetic data of sorption were well described by the pseudo-second-order model, indicating the sorption of the three antibiotics onto GAC involving valency forces through sharing or exchange of electrons between sorbent and sorbate. The Weber- Morris intraparticle diffusion model and Boyd kinetic model proved the main controlling step for the adsorption process was diffusion through the boundary layer. Using the adsorption equilibrium constant obtained from Langmuir isotherm, the thermodynamic parameter Δ*G*^0^ was calculated to tell the spontaneity of the adsorption reaction. The values of Δ*H*^0^ and Δ*S*^0^ were also obtained from a slope and intercept of the relationship between ln*K* and reaction temperature. A negative value of Δ*G*^0^ and a Negative value of Δ*H*^0^ confirmed the spontaneous and exothermic nature of the adsorption process. In conclusion, GAC could be employed as an environmentally friendly adsorbent for the removal of antibiotics from water and wastewater.

## Figures and Tables

**Figure 1 molecules-27-07980-f001:**
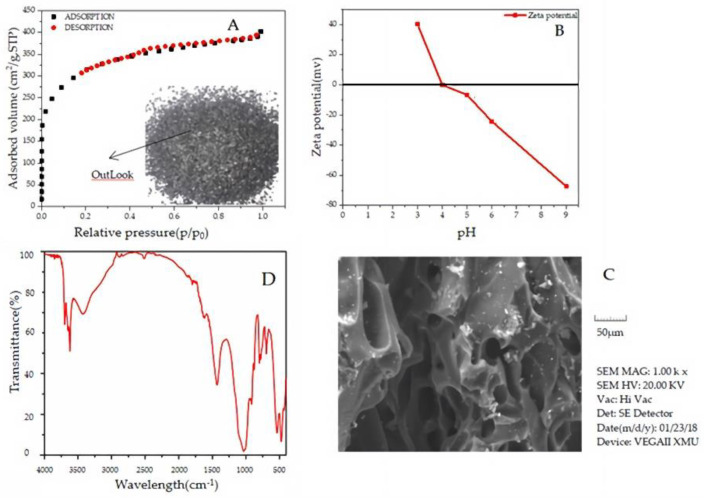
(**A**): Nitrogen (N_2_) adsorption isotherm (**B**): zeta potential of the GAC. (**C**): SEM image (**D**): FTIR image.

**Figure 2 molecules-27-07980-f002:**
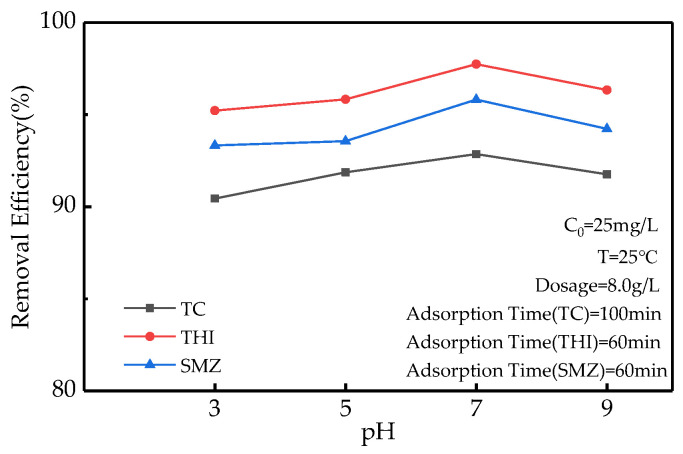
The relationship between pH and adsorption efficiency.

**Figure 3 molecules-27-07980-f003:**
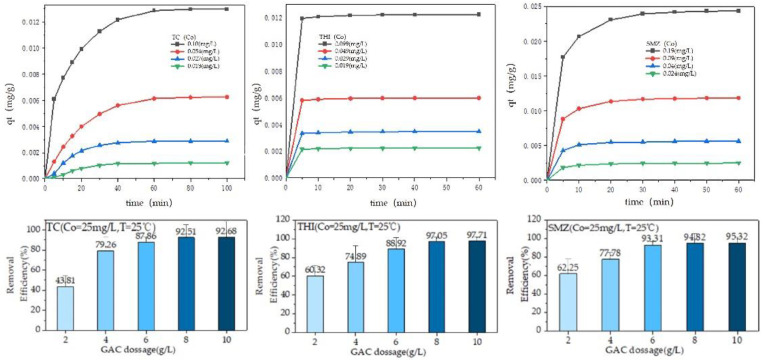
Influence of contact time on the adsorption of the three antibiotics by GAC under different initial concentrations.

**Figure 4 molecules-27-07980-f004:**
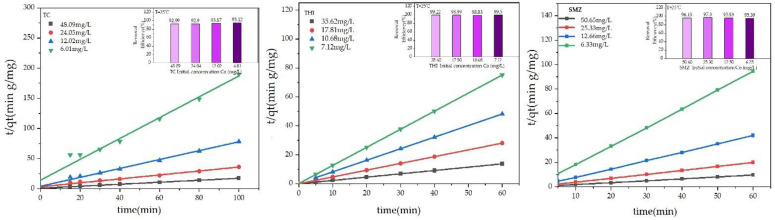
The pseudo-second-order kinetics for the adsorption of the three antibiotics on the GAC at various initial concentrations.

**Figure 5 molecules-27-07980-f005:**
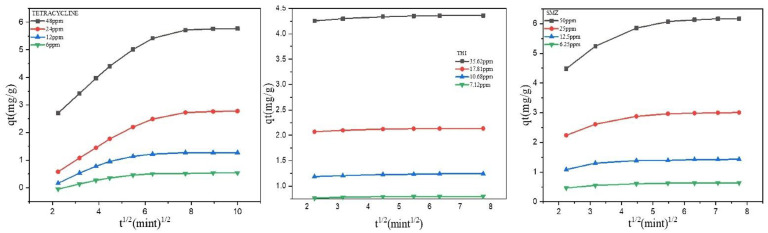
Intraparticle diffusion plots for the antibiotics sorption on the GAC.

**Figure 6 molecules-27-07980-f006:**
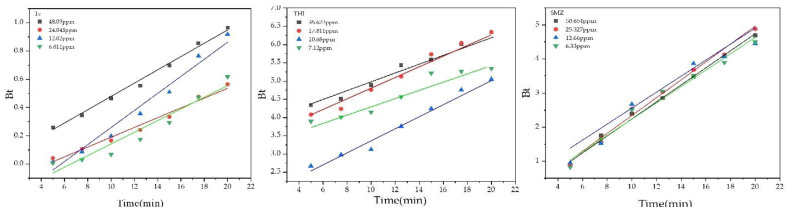
Plots of Bod kinetic model.

**Figure 7 molecules-27-07980-f007:**
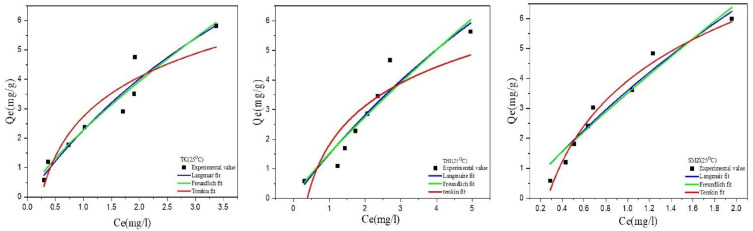
Plots of adsorption isotherms for the sorption of antibiotics on the GAC.

**Figure 8 molecules-27-07980-f008:**
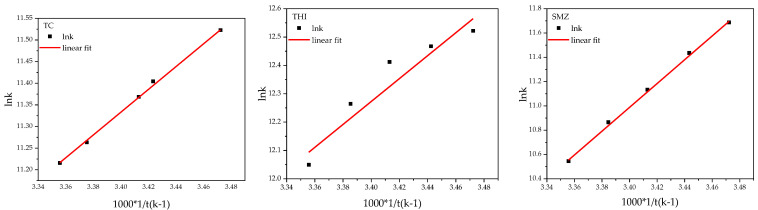
Plots of thermodynamics.

**Figure 9 molecules-27-07980-f009:**
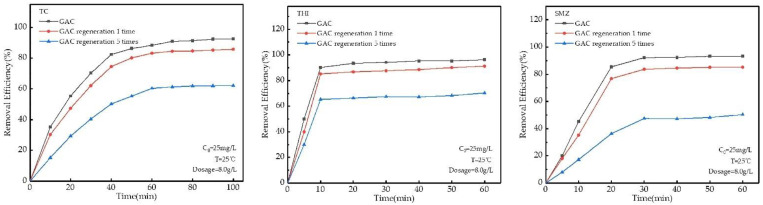
Relationship between GAC regeneration adsorption efficiency and time.

**Table 1 molecules-27-07980-t001:** Shows the results of the BET.

BET	Specific Surface Area	Low Pressure (*p*/*p*_0_ < 0.1) Adsorption Capacity	Hysteresis Loop (*p*/*p*_0_ = 0.2)	Total Pore Volume	Micro Pore Volume
	1059.011 m^2^/g	increased	closed	0.625 cm^3^/g	0.488 cm^3^/g
		micropores	mesoporous		

**Table 3 molecules-27-07980-t003:** Intraparticle diffusion kinetic model parameters at 25 °C.

Intra-Particle Diffusion Model							
Sorbate	*C*_0_ (mg/L)	*K_id_*_1_ (mg/g min^1/2^)	*I*_1_ (mg/g)	*R* ^2^	*K_id_*_2_ (mg/g min^1/2^)	*I*_2_ (mg/g)	*R* ^2^
TC	6.01	0.13	−0.3	0.94	0.01	0.44	0.83
	12.02	0.25	−0.29	0.93	0.01	1.10	0.58
	24.04	0.47	−0.4	0.99	0.10	1.84	0.77
	48.09	0.66	1.31	0.98	0.13	4.58	0.77
THI	7.12	0.00	0.75	0.70	-----	----	---
	10.68	0.01	1.17	0.85	-----	----	---
	17.81	0.01	2.06	0.75	-----	----	---
	35.62	0.01	4.23	0.79	-----	----	---
SMZ	6.33	0.06	0.33	0.88	0.00	0.56	0.82
	12.66	0.13	0.82	0.76	0.01	1.32	0.86
	25.32	0.28	1.65	0.92	0.03	2.73	0.78
	50.65	0.6	3.21	0.94	0.09	5.49	0.76

**Table 4 molecules-27-07980-t004:** The Boyd kinetic model parameters at 25 °C.

		Boyd Plot	
Sorbate	*C*_0_ (mg/L)	Intercept	*R* ^2^
TC	6.01	−0.026	0.92
	12.02	−0.344	0.97
	24.04	−0.155	0.98
	48.09	0.008	0.99
THI	7.12	2.640	1.00
	10.68	1.700	0.97
	17.81	2.870	0.96
	35.62	3.220	0.94
SMZ	6.33	−0.168	0.97
	12.66	0.206	0.89
	25.32	−0.285	0.99
	50.65	−0.262	0.99

**Table 5 molecules-27-07980-t005:** Parameters of the isotherm models describing the sorption of antibiotics on GAC at 25 °C.

Sorbate	Langmuir			Freundlich			Temkin		
	*q_m_* (mg/g)	*K_L_* (L/mg)	*R* ^2^	*K_F_* (mg/L^(1−1/n)^ g)	n	*R* ^2^	*B*	*K_T_*	*R* ^2^
TC	17.02	0.154	0.93	2.28	1.2	0.93	1.944	4.093	0.88
THI	30.42	0.530	0.92	12.17	1.1	0.91	-----	----	----
SMZ	26.77	0.155	0.95	3.51	1.1	0.94	2.930	3.800	0.97

**Table 6 molecules-27-07980-t006:** Separation factor.

Value of *R_L_*	Adsorption Nature
0 < *R_L_* < 1	Favorable
*R_L_* = 0	Irreversible
*R_L_* = 1	Linear
*R_L_* > 1	Unfavorable

**Table 7 molecules-27-07980-t007:** Thermodynamics parameters for the adsorption of three antibiotics on GAC.

Sorbate	*T* ^0^ *C*	ln*K*	Δ*G*^0^ (kJ/mol)	Δ*H*^0^ (kJ/mol)	Δ*S*^0^ (kJ/mol K)
TC	15	11.522	−26.97		
	20	11.368	−27.67	−21.86	19.17
	25	11.215	−27.89		
THI	15	12.521	−29.98		
	20	12.412	−30.23	−33.58	−12.22
	25	12.049	−29.85		
SMZ	15	11.687	−27.98		
	20	11.132	−27.11	−81.39	−185.56
	25	10.545	−26.12		

**Table 8 molecules-27-07980-t008:** Physicochemical properties of the three antibiotics.

Property	Tetracycline (TC)	Thiamphenicol (THI)	Sulfamethoxazole (SMZ)
Molecular formula & Chemical Structure	C_22_H_24_N_2_O_8_ 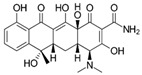	C_12_H_15_CL_2_NO_5_S 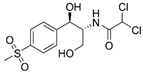	C_10_H_11_N_3_O_3_S 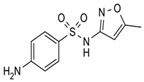
Molar mass	444.43	356.22	253.28
Solubility (25 °C)	1700 mg/L	2270 mg/L	459 mg/L

## Data Availability

Not applicable.

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
