# Peer review of "Adsorption of Toxic Tetracycline, Thiamphenicol and Sulfamethoxazole by a Granular Activated Carbon (GAC) under Different Conditions"

_molecules, 2022, doi:10.3390/molecules27227980_

Round 1

Reviewer 1 Report (New Reviewer)

This manuscript deals with adsorptive removal of antibiotics, Tetracycline (TC), Thiamphenicol (THI), and Sulfamethoxazole (SMZ), from their aqueous solution by activated carbon in granular form (GAC). The adsorption kinetics, isotherm, and thermodynamics of the three different antibiotics were analyzed and discussed. After reading this manuscript carefully, several crucial and critical issues should be considered before considering this manuscript for publication in Molecules.

1.         In general, the manuscript should be proofread. They are many typos, wrong spelling, and unclear sentences.

2.         Introduction; Many statements in Introduction are incorrect and misleading. It is the matter of fact that recent developments of the adsorption of organic compounds and antibiotics have discovered many interesting findings. Authors just omitted those new findings and focused on old references published in average more than 10 years ago.   

3.         Line 62-63; Many advanced oxidation processes are inexpensive (see for instance; Heliyon 2022, 8, e10137; Chem. Eng. J. 2022, 431, 133916; J. Colloid Interface Sci. 2022, 607, 568–583; Chemosphere 2020, 253, 126595; Chemosphere 2019, 227, 198–206; Water 2020, 12, 1–50). In fact, the cost to produce activated carbon in this study is not presented, but anyway the production cost of activated carbon should be more expensive as compared with naturally available agricultural waste.

4.         Line 69; What is the biologically process in adsorption?

5.         Line 69-70; There is no references for the use of adsorption in the removal of 30 antibiotics.

6.         Line 72; There are many reports highlighting that adsorption of organic compounds relies on hydrogen bonding and electrostatic interactions, rather than the surface area, pore size, and pore volume.

7.         Line 77; see comment #3.

8.         Line 100: It is very strange that the preparation of granulated activated carbon is not explained in this study. Claim on the low cost should be supported with an analysis of production cost.

9.         Line 101: It is important to note that adsorption process is based on interactions between adsorbate (in this case the antibiotics) and the functional groups on the adsorbent surface. In this case, the chemical structures of the antibiotics should be presented. In addition, there are many simulation models have been explored to describe the interactions.

10.     Line 118: What is the reason for the high GAC dosages (from 2 to 8 g/L) in this study, as usually the adsorbent dosage is less than 1 g/L to remove the ppm levels of organic compounds. With that adsorbent over dosages, the adsorption efficiency is always claimed to be close to 100%.

11.     Line 128-129: How filter liquor can be used to determine the residual concentrations of the three antibiotics? In adsorption studies, the use of filter is not recommended, as filter papers could adsorb the target adsorbate, resulting in inaccurate analysis.   

12.     Line 156-158: In fact, the UV-vis spectrophotometer is used to determine the concentration of antibiotics. However, it is very strange that there is no absorption spectrum is presented. More importantly, Tetracycline (TC), Thiamphenicol (THI), and Sulfamethoxazole (SMZ) have absorption spectral peaks at 370 nm, 230 nm, and 260 nm. In this regard, the absorption should be strongly affected by Rayleigh scattering of the leftover GAC adsorbent, and the effect of scattering should be removed before analysis.

13.     Figure 1D. Adsorption process should be analyzed based on FTIR spectra of GAC before and after adsorption. It is also encouraged to show XRD patterns of GAC before and after adsorption, anticipating the structural changes due to the adsorption.

14.     Figure 2. The plot is pH in the range of 3-9, inconsistent with the description in Line 152 (pH 1-9). It is also crucial to evaluate the pKa of Tetracycline (TC), Thiamphenicol (THI), and Sulfamethoxazole (SMZ). In this sense, the ionic state of the antibiotics may change at certain pH, so that their electrostatic interaction with the GAC surface is obviously modified. This should be discussed together with the net surface charge of GAC.

15.     Figure 4 and Table 4: The plots of pseudo–second-order kinetics are presented, but those of pseudo–first-order kinetics are not, though their parameters are given in Table 4.

16.     Line 271-277: Interpretation of the Weber-Morris plot should be corrected. Accordingly, the plot should be interpolated to the origin.

17.     Figure 5B: Authors should explain the flat graph of Thiamphenicol (THI).

18.     Figure 6: It is very strange that the Boyd plots only contain 3 or 4 data points. Basically, the Boyd plots are based on the same data used in the Weber-Morris plots. What is the reason to remove some kinetic data in the Boyd analysis?

19.     Line 199-200: Interpretation of the Boyd plots should be corrected, and the plots should be interpolated to the origin.

20.     Section 3.6: The discussion on adsorption isotherm is very poor. Authors should elaborate it in detail.

21.     Line 331: The separation factor is used to explain the separation of multiple dyes from binary solution using adsorption process, not on single solution.

22.     Table 1: This table is meaningless.

23.     Line 343-356: What is the basis of k calculation? Authors may refer

J. Mol. Liq. 2019, 273, 425–434; Sep. Purif. Technol. 2008, 61, 229–242

Many papers have discussed the use of this Gibbs equation in adsorption in great detail.

24.     Figure 8: It is difficult to obtain a good liner regression and justify the parameters just based on 3 data points.

25.     Table 8: Authors should explain why Tetracycline (TC) gives positive entropy, while Thiamphenicol (THI) and Sulfamethoxazole (SMZ) give negative entropy. If the latter is true, it means that the GAC surface is originally unstable.

26.     Line 276-379; It is repetitive of line 353-355.

27.     Adsorption of Tetracycline (TC), Thiamphenicol (THI), and Sulfamethoxazole (SMZ) is not reported for the first time in this study. In fact, the adsorption of these antibiotics has been reported in literature. Authors should discuss the scientific achievement, such as the maximum adsorption capacity of the antibiotics in this study with those reported in literature.

Author Response

  1. In general, the manuscript should be proofread. They are many typos, wrong spelling, and unclear sentences.

Dear reviewer, the article has been checked for spelling problems sentence by sentence and grammatical errors corrected.

  1. Introduction; Many statements in Introduction are incorrect and misleading. It is the matter of fact that recent developments of the adsorption of organic compounds and antibiotics have discovered many interesting findings. Authors just omitted those new findings and focused on old references published in average more than 10 years ago.

Dear Reviewer:

The introduction of this manuscript has been revised according to the questions you raised below, please review it.

  1. Line 62-63; Many advanced oxidation processes are inexpensive (see for instance; Heliyon 2022, 8, e10137; Chem. Eng. J. 2022, 431, 133916; J. Colloid Interface Sci. 2022, 607, 568–583; Chemosphere 2020, 253, 126595; Chemosphere 2019, 227, 198–206; Water 2020, 12, 1–50). In fact, the cost to produce activated carbon in this study is not presented, but anyway the production cost of activated carbon should be more expensive as compared with naturally available agricultural waste.

I am very grateful to the reviewer for informing me of the latest advanced oxidation process, and I have carefully read the article you recommended. And delete inappropriate statements in the introduction.

The manuscript was revised as follows:

“It is important to dispose residues of antibiotic before  discharging wastewater in environment. There is an urgent need for case studies to provide cheap solution for the elimination of antibiotics. Sera Budi Verinda et al. used ozonation to remove ciprofloxacin in wastewater, and the removal rate reached 83.5%[14];Shang,K.F. et al. indicates that the combination of DBD plasma and PMS/PDS is an efficient pretreatment method for bio-treatment of refractory SMX[15].

  1. Line 69; What is the biologically process in adsorption?

Dear Reviewer,

I misrepresented this sentence, what I meant was that the adsorbent should be a biomass adsorbent such as agricultural waste, which is environmentally friendly and economical.

The manuscript was revised as follows:

“The adsorbent should be a biomass adsorbent such as agricultural waste, which is environmentally friendly and economical.[17].

  1. Line 69-70; There is no references for the use of adsorption in the removal of 30 antibiotics.

Dear Reviewer

The manuscript was revised as follows:

“Its application to the elimination of antibiotics for approximately 30 different compounds have been reported so far.[19]

  • Homem,V.;Santos, L. Degradation and removal methods of antibiotics from aqueous matrices-a review[J]. Journal of Environmental Management 2011,92(10), 2304-2347.

  1. Line 72; There are many reports highlighting that adsorption of organic compounds relies on hydrogen bonding and electrostatic interactions, rather than the surface area, pore size, and pore volume.

Dear Reviewer

The manuscript was revised as follows:

“The performance of adsorption processes is largely influenced by  the hydrogen bonding and electrostatic interactions.

  1. Line 77; see comment #3.

Dear Reviewer

The manuscript was revised as follows:

“Compared with traditional adsorption materials, activated carbon has excellent porosity, large specific surface area, low cost and environmental friendliness, and is reported to be an effective adsorbent for eliminating trace pollutants.”

  1. Line 100: It is very strange that the preparation of granulated activated carbon is not explained in this study. Claim on the low cost should be supported with an analysis of production cost.

Dear Reviewer:

The manuscript was revised as follows:

“2.1. GAC Preparation

The Granular activated carbon used in these experiments was prepared by using corn Stover that was collect from Shaanxi agriculture Technology Company (Xian china). First this corn Stover was put 1 month for air drying. After that it was chop into small pieces of length nearly 5cm and positioned inside electrically operating container resistance furnace (LNB4-13Y; Haozhuang Co. Ltd., Shanghai, China). Under nitrogen atmosphere where pieces of cornstalk were heated at 500oC for 2 h and then 700oC for 2 h.

Physical activation management was done in an activation furnace (HHL-1; Huatong Co. Ltd., Henan, China) through superheated steam at very elevated temperatures (600oC, 2.0 MPa) for a duration of 2 hours to obtain the AC. The AC was also cleansed using 1.0M HCl-HF (1:1, vol/vol) solution thrice and with ultrapure water a few times till the 7.0 pH value was obtained. After washing, drying of AC was then done at a temperature of 100oC for a duration of 15 hours, and then crushed and pushed through a nylon mesh having a entrance of 5.0 mm. Finally, the AC’s morphology was granular. “

  1. Line 101: It is important to note that adsorption process is based on interactions between adsorbate (in this case the antibiotics) and the functional groups on the adsorbent surface. In this case, the chemical structures of the antibiotics should be presented. In addition, there are many simulation models have been explored to describe the interactions.

Dear Reviewer:

The chemical structures of the three antibiotics have been supplemented in Table 1:

Property                      

Tetracycline (TC)

Thiamphenicol (THI)

Sulfamethoxazole (SMZ)

Molecular formula & Chemical Structure

C22H24N2O8

C12H15CL2NO5S

C10H11N3O3S

Molar mass

444.43

356.22

253.28

Solubility (25 ℃)

1700mg/L

2270mg/L

459mg/L

  1. Line 118: What is the reason for the high GAC dosages (from 2 to 8 g/L) in this study, as usually the adsorbent dosage is less than 1 g/L to remove the ppm levels of organic compounds. With that adsorbent over dosages, the adsorption efficiency is always claimed to be close to 100%.

Dear Reviewer:

The GAC dosage gradient experiment found that when the GAC concentration was 2-8g/L, the removal efficiency of antibiotics was the best.

  1. Line 128-129: How filter liquor can be used to determine the residual concentrations of the three antibiotics? In adsorption studies, the use of filter is not recommended, as filter papers could adsorb the target adsorbate, resulting in inaccurate analysis.   

Dear Reviewer:

This article uses 0.45um membrane filtration to determine the antibiotic concentration. Your suggestion is very good. In the future, it will be changed to 8000 rpm centrifugation for 10 minutes and the supernatant will be measured.

  1. Line 156-158: In fact, the UV-vis spectrophotometer is used to determine the concentration of antibiotics. However, it is very strange that there is no absorption spectrum is presented. More importantly, Tetracycline (TC), Thiamphenicol (THI), and Sulfamethoxazole (SMZ) have absorption spectral peaks at 370 nm, 230 nm, and 260 nm. In this regard, the absorption should be strongly affected by Rayleigh scattering of the leftover GAC adsorbent, and the effect of scattering should be removed before analysis.

Dear Reviewer:

The experimental result obtained by the UV spectrophotometer used in this manuscript is the absorbance A, not the absorption spectrum. Due to the limitation of the instrument model, the removal of the Rayleigh scattering effect proposed by you cannot be satisfied temporarily.

  1. Figure 1D. Adsorption process should be analyzed based on FTIR spectra of GAC before and after adsorption. It is also encouraged to show XRD patterns of GAC before and after adsorption, anticipating the structural changes due to the adsorption.

Dear reviewer:

I think the focus of this manuscript is on the removal of the four antibiotics by granular activated carbon, rather than the characterization of the activated carbon itself, so the analysis of the infrared spectrum after the removal of granular activated carbon is somewhat deviated from the central meaning of the article.

  1. Figure 2. The plot is pH in the range of 3-9, inconsistent with the description in Line 152 (pH 1-9). It is also crucial to evaluate the pKa of Tetracycline (TC), Thiamphenicol (THI), and Sulfamethoxazole (SMZ). In this sense, the ionic state of the antibiotics may change at certain pH, so that their electrostatic interaction with the GAC surface is obviously modified. This should be discussed together with the net surface charge of GAC.

Dear reviewer:

The manuscript was revised as follows:

“For adjusting the pH value of solution from 3 to 9 HCl or NaOH was used.”

  1. Figure 4 and Table 4: The plots of pseudo–second-order kinetics are presented, but those of pseudo–first-order kinetics are not, though their parameters are given in Table 4.

Dear Reviewer:

It is not discussed in the manuscript due to the suboptimal fit of the first-order kinetic curve.

  1. Line 271-277: Interpretation of the Weber-Morris plot should be corrected. Accordingly, the plot should be interpolated to the origin.
  2. Figure 5B: Authors should explain the flat graph of Thiamphenicol (THI).

Dear Reviewer:

I will simply answer your questions 16 and 17 here:

If the plot between qt versus t1/2 would be linear then intraparticle diffusion take place. When the plot also passes through origin (I=0) then the rate limiting is con-trolled by intraparticle diffusion. If the plots showed deviation from linearity, then it indicates the effect of boundary layer. Fig. 5 represents the plot of intraparticle for three antibiotics. This was observed that during whole time plots linear portion cannot pass through origin, indicating that both boundary layer and intraparticle diffusion take place during the adsorption of antibiotics on GAC.

Therefore, it is more appropriate for my task to fit the curve without going through the origin, and the above explanation also explains the THI in Figure 5B.

  1. Figure 6: It is very strange that the Boyd plots only contain 3 or 4 data points. Basically, the Boyd plots are based on the same data used in the Weber-Morris plots. What is the reason to remove some kinetic data in the Boyd analysis?

Dear Reviewer:

Figure 6 does not delete kinetic data, but selects data at four time points of 5min, 10min, 15min, and 20min for fitting.

  1. Line 199-200: Interpretation of the Boyd plots should be corrected, and the plots should be interpolated to the origin.

Dear Reviewer:

As for the Boyd kinetic model, if the plot of Bt against t is linear and through the origin, it suggests the fact that intraparticle diffusion is controlling the process of mass transfer. When the plot can be seen as nonlinear or linear but does not go through the origin, it means that the sorption rate is controlled through film diffusion. Fig.6 illus-trates the Boyd plots for the three antibiotics on the GAC sample. The fact that Boyd plots were linear even through they were unable to pass through the origin was no-ticed. These results indicate the information regarding the fact that the major control-ling process required for adsorption procedure was nonetheless diffusion with the layer at border.

So I don't think it's appropriate to draw the fitted curve past the origin.

  1. Section 3.6: The discussion on adsorption isotherm is very poor. Authors should elaborate it in detail.

Dear Reviewer:

The supplementary content is as follows:

Adsorption process can be defined as the process where mass transfer of adsorb-ate at the boundary layer in-between the solid adsorbent and liquid phase takes place. The adsorption isotherm is defined as the equilibrium relationship between solid ad-sorbent and adsorbate at constant temperature. For example: the ratio between amount of adsorbate absorbed on the solid and the remaining amount left in aqueous solution at equilibrium.  There are several adsorption isotherms models like Langmuir, Freun-dlich, Temkin.

Experimental data can be fitted using these models to examine the suitability of model. The information obtained from these models can be used for designing the ad-sorption process. The parameters of adsorption isotherm normally estimated the sorp-tion ability of several adsorbent for specific adsorbate with predetermine reaction con-dition. The performance of the sorption process depends not only on the rate at which mass transfer occurs but also on the sorbent–sorbate equilibrium concentration [34]Three different  isotherm models, called the, Langmuir [35], Freundlich [36], and Temkin [20,21]were used. The mathematical formula of these isotherm models were shown as follows:

  1. Line 331: The separation factor is used to explain the separation of multiple dyes from binary solution using adsorption process, not on single solution.

Dear Reviewer:

Information on segregation factors has been removed from the manuscript.

  1. Table 1: This table is meaningless.

Dear Reviewer:

The chemical structure diagram you suggested is listed in Table 1, so that the properties of the three antibiotics can be more intuitively displayed.

Property                      

Tetracycline (TC)

Thiamphenicol (THI)

Sulfamethoxazole (SMZ)

Molecular formula & Chemical Structure

C22H24N2O8

C12H15CL2NO5S

C10H11N3O3S

Molar mass

444.43

356.22

253.28

Solubility (25 ℃)

1700mg/L

2270mg/L

459mg/L

  1. Line 343-356: What is the basis of k calculation? Authors may refer
  2. Mol. Liq. 2019, 273, 425–434; Sep. Purif. Technol. 2008, 61, 229–242

Many papers have discussed the use of this Gibbs equation in adsorption in great detail.

Dear Reviewer:

The calculation of the K value is based on the most primitive and classic Gibbs equation: .

“A critical review of the estimation of the thermodynamic parameters on adsorption equilibria. Wrong use of equilibrium constant in the Van't Hoof equation for calculation of thermodynamic parameters of adsorption” I have yet to learn about the use of the latest Gibbs equation in the article in the adsorption process, thanks for the advice.

  1. Figure 8: It is difficult to obtain a good liner regression and justify the parameters just based on 3 data points.

  1. Table 8: Authors should explain why Tetracycline (TC) gives positive entropy, while Thiamphenicol (THI) and Sulfamethoxazole (SMZ) give negative entropy. If the latter is true, it means that the GAC surface is originally unstable.

Dear Reviewer:

The negative value of ΔS0 suggests that the adsorption process is enthalpy driven.

The negative value of the entropy change (ΔS0) also means that the disorder of the solid/liquid interface decreases during the adsorption process, resulting in the escape of adsorbed ion/molecules from the solid. Therefore, the amount of adsorbate adsorbed will decrease.

  1. Line 376-379; It is repetitive of line 353-355.

Dear Reviewer:

The article has deleted lines 376-379

  1. Adsorption of Tetracycline (TC), Thiamphenicol (THI), and Sulfamethoxazole (SMZ) is not reported for the first time in this study. In fact, the adsorption of these antibiotics has been reported in literature. Authors should discuss the scientific achievement, such as the maximum adsorption capacity of the antibiotics in this study with those reported in literature.

Dear Reviewer:

The equilibrium adsorption data of TC, THI, and SMZ were good expressed by Lang-muir isotherm model, and maximum adsorption capacity were 17.02, 30.40, and 26.77mg/g respectively.

Reviewer 2 Report (New Reviewer)

Review on:

Adsorption of toxic tetracycline, thiamphenicol and sulfameth-oxazole by a granular activated carbon (GAC) under different conditions

The research and the article have been well prepared. Data taken is sufficient for a comprehensive study in the adsorption. However, some improvement is required to increased the quality by addressing some issues below:

1.     FTIR spectra should be equipped with a description of the molecular vibrations at each peak. FTIR spectra in Fig.1: an intense peak at around 3600 – 3700 cm-1 is definitely not -OH vibration. Authors shall explain all of peaks reveals within, including a very intense peak at 500 cm-1,

2.     Each figure and table should be referred in text, Table 2 is not referred in text,

3.     Please be aware on writing constant equilibrium with K. It should not k, as written in equation (15). Author took the values of equilibrium thermodynamics constant from Langmuir constant, which is written as KL , so it is mandatory to be consistent in writing every sign in this paper.

Author Response

The research and the article have been well prepared. Data taken is sufficient for a comprehensive study in the adsorption. However, some improvement is required to increased the quality by addressing some issues below:

  1. FTIR spectra should be equipped with a description of the molecular vibrations at each peak. FTIR spectra in Fig.1: an intense peak at around 3600 – 3700 cm-1 is definitely not -OH vibration. Authors shall explain all of peaks reveals within, including a very intense peak at 500 cm-1,

Dear Reviewer:

3600-3700cm-1 is O-H stretching vibration peak,  Since the activated carbon used in this manuscript was prepared from straw, the peak at 500cm-1 may be ν(P-P)

  1. Each figure and table should be referred in text, Table 2 is not referred in text,

Dear Reviewer:

The details of BET are shown in Table 2:These results demonstrated that GAC was of high specific surface area and porous structure.

  1. Please be aware on writing constant equilibrium with K. It should not k, as written in equation (15). Author took the values of equilibrium thermodynamics constant from Langmuir constant, which is written as KL , so it is mandatory to be consistent in writing every sign in this paper.

Dear Reviewer:

The manuscript was revised as follows:

Round 2

Reviewer 1 Report (New Reviewer)

Title:

Adsorption of toxic tetracycline, thiamphenicol and sulfameth-oxazole by a granular activated carbon (GAC) under different conditions

My comments:

This manuscript deals with adsorption of three different antibiotics (Tetracycline, Thiamphenicol, and Sulfamethoxazole) using granulated activated carbon. The adsorption kinetics, diffusion, isotherm, and thermodynamics of the antibiotics have been evaluated. The manuscript has been improved from its original form. However, several minor corrections and suggestions should be considered before accepting this manuscript for publication.

1.         Line 160: It should be Barrett-Joyner-Halenda (BJH) method.

2.      Line 169: It is mentioned that the UV-vis spectrometer is used to measure the concentration of the antibiotics. Authors should show the absorption spectra of the solution of the antibiotics before and after adsorption.

3.         Figure 1: The values in the Figure 1(B) should be removed, as the y-axis shows the values.

4.         Line 260-269: Here, authors discuss about the data presented in Table 3. It is important to highlight the trend of k2 and Qe,calc as a function of C0.

5.         Line 282-283: The sentence is unclear.

6.         Table 4: It is essential to describe the values of I. Why I1 values are negative for TC, what does it mean?

7.         Figure 6: The Boyd diffusion model is usually plotted based on the same kinetic data to those used in the intraparticle diffusion model. In this regard, authors should use all the data used in the intraparticle diffusion plots (Figure 5: there are 6-9 data points). Is there any reason for just using 3-4 data points only?

8.         Table 8: The unit of ΔS should be kJ/mol.K to be uniform with ΔG and ΔH. It is also important to explain the different signs of ΔS for TC, THI, and SMZ.

Author Response

Reply to Reviewer Comments

  1. Line 160: It should be Barrett-Joyner-Halenda (BJH) method.

Dear Reviewer:

The manuscript was revised as follows:

The Brunauer-Emmett-Teller (BET) method was used to measure surface area and Bar-rett-Joyner-Halenda (BJH) method was used to measure pore size distribution.

  1. Line 169: It is mentioned that the UV-vis spectrometer is used to measure the concentration of the antibiotics. Authors should show the absorption spectra of the solution of the antibiotics before and after adsorption.

Dear Reviewer:

The concentration of antibiotics was detected by ultraviolet spectrophotometry, taking the absorbance A as the ordinate and the concentration c as the abscissa to obtain the standard curve. Substitute the absorbance A of the antibiotic solution before and after activated carbon adsorption into the standard curve equation to obtain its concentration. Therefore, the absorption spectra of the solution of the antibiotics before and after adsorption,you speak of cannot be represented in the manuscript.

  1. Figure 1: The values in the Figure 1(B) should be removed, as the y-axis shows the values.

Dear Reviewer:

Figure 1(B) is modified as follows:

  1. Line 260-269: Here, authors discuss about the data presented in Table 3. It is important to highlight the trend of k2 and Qe,calc as a function of C0.

Dear Reviewer:

The manuscript was revised as follows:

These results indicate that rate of adsorption on GAC is controlled by chemisorption, while valency forces with exchange or there might be a sharing of the electrons in-between these four antibiotics and the GAC, according to Table 3 primary initial adsorption rate K2q2 e,cal gradually increased with the increase in the concentration of primary four antibiotics, indicating the fact that the greater value of concentration en-hanced driving forces that can basically help to overcome the barrier of mass trans-fer-resistance in-between the phases of solid and liquid. 

  1. Line 282-283: The sentence is unclear.

Dear Reviewer:

The manuscript was revised as follows:

                                               (5)

Where

qt(mg/g) = the removal amount at time t and reaction equilibrium

Kid (mg/g⋅min1/2)= the particle diffusion rate constant

  1. Table 4: It is essential to describe the values of I. Why I1 values are negative for TC, what does it mean?

Dear Reviewer:

The manuscript was revised as follows:

In the Table 4, the rate constant and intercept values were shown. The intercepts C of the straight lines fitted by TC, THI, and SMZ are not 0, indicating that internal diffusion is not the only step controlling the removal of antibiotics by GAC, and the adsorption rate should be controlled by both external diffusion and intraparticle diffu-sion.

  1. Figure 6: The Boyd diffusion model is usually plotted based on the same kinetic data to those used in the intraparticle diffusion model. In this regard, authors should use all the data used in the intraparticle diffusion plots (Figure 5: there are 6-9 data points). Is there any reason for just using 3-4 data points only?

Dear Reviewer:

The manuscript was revised as follows:

  1. Table 8: The unit of ΔS should be kJ/mol.K to be uniform with ΔG and ΔH. It is also important to explain the different signs of ΔS for TC, THI, and SMZ.

Dear Reviewer:

The manuscript was revised as follows:

Sorbate

ToC

Lnk

ΔG°(Kj/mol)

ΔH° (Kj/mol)

ΔS° (Kj/mol. K)

TC

15

11.522

-26.97

20

11.368

-27.67

-21.86

19.17

25

11.215

-27.89

THI

15

12.521

-29.98

20

12.412

-30.23

-33.58

-12.22

25

12.049

-29.85

SMZ

15

11.687

-27.98

20

11.132

-27.11

-81.39

-185.56

25

10.545

-26.12

This manuscript is a resubmission of an earlier submission. The following is a list of the peer review reports and author responses from that submission.

Round 1

Author Response

Response to Reviewer 1 Comments

Point 1:revise the sentence

Response 1:When antibiotics are used, and the organisms cannot fully absorb the antibiotics, the antibiotics are released into the environment in an active form.

Point 2:if antibiotic could give adverse effect when contacted to it, them what would happen if we consume the antibiotic as medicine?

Response 2:When antibiotics are used by humans as medicines when the dose exceeds a certain amount, human allergic reactions will occur, and when the dose is seriously exceeded, the human immune system and nervous system may even be damaged.

Point 3:but we still using it as a medicine

Response 3:When the dose of antibiotics is within the acceptable range of the human body, it can treat human diseases.

Point 4:but we still using it as a medicine

Response 4:When the dose of antibiotics is within the acceptable range of the human body, it can treat human diseases.

Point 5:not solve the issue actually, just mitigate the antibiotic from water bodies to other place

Response 5:There is indeed no fundamental solution to the transfer of antibiotics from water bodies to other places, but from another perspective, antibiotics are adsorbed to active substances, which purify the water body on the one hand and adsorb the antibiotics on the other hand. Said to be more easily degraded.

Point 6:why too low

Response 6:When the temperature is higher than 25 degrees Celsius, the adsorption efficiency of TC and GAC will decrease.

Point 7:is it practical?

Response 7:Yes.

Point 8:No repetition for the experimental work

Response 8:Thanks for your suggestion, we will make further changes in the next experiment.

Point 9:include error bar

Response 9:Dear reviewer, because the experiment did not carry out parallel experiments, it is impossible to add error bars to the icons. This is also the biggest shortcoming of our experiment. We gladly accept your suggestion and will be in future experiments. Supplementary corrections.

Point 10:

overall comment for result and discussion part

1. No error analysis

2. The finding is not discussed critically. No support from previous research.

3. The author should discuss the effect of different characteristic of the antibiotic towards its removal.

4. It is more interesting if the antibiotic is tested in mixture -not just individual

Response 10: 

Dear Reviewer:

I will reply to your suggestion in the following points:

1. Not doing parallel experiments is a mistake in our exploration this time, and we will make supplementary corrections in the next experiments;

2. There are few supporting materials related to the activated carbon adsorption of antibiotics because this is also theoretical support provided by this paper for the adsorption of antibiotics on activated carbon;

3.4 Your two suggestions are very good and provide a new way of thinking for our research. We will explore these two points in more depth in the next experiments.

Author Response

  1. The novelty of the study must to be presented in the Introduction part.

Dear Reviewers:

The last paragraph of the introduction is as follows:

At present, there are very few related research on antibiotics in wastewater through activated carbon,In this paper, three different kinds of antibiotics including tetracycline, thiamphenicol and sulfamethoxazole were selected as the target pollutants and a kind of activated carbon in granular form was used as the adsorbent. The adsorption capacity and mechanism were studied. The purpose of this paper was to evaluate the removal efficiency of adsorption technology on the treatement of antibiotics-loading wastewater and to promote the application of activated carbon in such a field.  

  1. Please add in the manuscript a description of the synthesis of GAC sample. The authors used the same synthesis conditions as Fan et al.? Or the synthesis conditions were modified?

Dear Reviewer:

The GAC preparation conditions used here are the same as Fan et al.

  1. I don’t have access to the manuscript of Fan and co-workers (reference 26). I have access only to the Abstract part in which I read that ‘’cornstalk was employed as a raw material to manufacture a kind of granular activated carbon (GAC), which was then used to remove four antibiotics, including isoniazid (IN), sulfamethoxazole (SMZ), thiamphenicol (THI), and doxycycline (DOX), from water’’. In the present study, the authors obtained a maximum adsorption capacities of 30.4 mg/g for THI and 26.77 mg/g for SMZ, while the research team of Yurui Fan shows based on Langmuir isotherm model a maximum adsorption capacities of 36.9301 mg/g for SMZ and 28.0637 mg/g for THI, respectively. The authors must to specify the originality and highlight why their results have an interest for the scientific community compared with the study presented by Fan et al.

Dear Reviewer:

Reference 26:Fan, Y.; Zheng, C. eds. Preparation of Granular Activated Carbon and Its Mechanism in the Removal of Isoniazid, Sulfamethoxazole, Thiamphenicol, and Doxycycline from Aqueous Solution. Environ. Eng. Sci. 2019. 36, 1027-1040.

In this paper, Fan et al. describe in detail the preparation process and characteristics of GAC, and the use of this specific GAC to degrade antibiotics in wastewater is a better application of the GAC prepared by Fan et al.

  1. Figure 1B should be improved. Consequently, I suggest to add the SEM image as Figure 1C. Also correct ph with pH (x-axis) .

Dear Reviewer: The chart is changed as follows

  1. I suggest to add FTIR of GAC sample.

Dear reviewer:

Fourier's infrared spectrum is a good way of representation. Due to the experimental cycle, we will add FTIR to the method of the next study. Thank you for your suggestion.

  1. The presentation of the Sections 3.2 and 3.3 must be improved. The information are not presented in a good manner. Consequently:

o Please include the graph corresponding to the effect of GAC dosage on the adsorption at Section 3.2 at the end of paragraph (Line 187). Please, express the optimization of adsorption parameter based on the adsorption capacity, mg/g as well (for the effect of GAC dosage on the adsorption);

Dear reviewer:

Section 3.2 changes as follows:

Fig 2 shows the removal efficiency versus GAC dosage. When the GAC dosage increased from 2 to 6 mg/g, all the removal efficiencies for the three antibiotics increased remarkably and then slightly from 6 to 10 mg/g. A possible reason was that with an increase in GAC dosage, the sorbent surface area, the number of sorption sites, and the contact area increased. In this study, considering the removal efficiencies and economic benefits, a GAC dosage of 4 mg/g was selected as an optimum dosage.

o Please increase the resolution of the graph which shows the effect of GAC dosage on the adsorption

o Figure 2: Please express qt in mg/g and not in mol/g (y-axis)

o Figure 2: Please express the initial concentration in mg/L and not in mol/l.

o At Section 2.2 the authors mentioned that: ‘’For GAC dosage, the amount of GAC ranged from 2 to 8 g/L….’’, but at Section 3.2 it is mentioned that: ‘’ When the GAC dosage increased from 2 to 6 g/L, all the removal efficiencies for the three 2 antibiotics increased remarkably and then slightly from 6.0 to 10 g/L.’’ Also, from the graphs it can be seen that this parameter was investigated between 2 – 10 g/L.

Dear reviewer:

When the GAC usage is 2-6g/L, the efficiency increases rapidly,

Remove efficiency growth slowly when 6-8g/L.

Removal efficiency is basically not increased when 8-10g/L.

Therefore, in Section 2.2, it is expressed as 2-8g/L as a reasonable amount.

  1. Add a new section for the effect of initial concentration。

Dear reviewer:

The effect of the initial concentration can be reflected in Figure 2, please refer to it.

  1. Please check the Figure 3. For the results of THI and SMZ, please express the initial concentration in mg/L.

  1. Add the fit of Pseudo first order kinetic model .

Dear reviewer:

I’m so sorry to tell you that the curve fitted by the first-order kinetic model formula is not ideal, so it is not described in the text. I think the question you raised is very correct, but in order to keep the length of the article and the graph within the number of words required by the journal, we give up A description of the first-order kinetic model is presented.

  1. Please check the y-axis: t/qt (min g/mg) instead of t/qt(mint.g/mg) and t/qt(minute.g/mg), respectivel

  1. Figure 6. The fit of Temkin isotherm is not presented in the case of THI Additionally, at the end of the study add:

o A comparison between GAC sample and other adsorbents presented in the literature for the removal of TC, THI, and SMZ

o An adsorption/desorption study in order to establish for how many cycles the GAC sample can be utilized for the removal of tetracycline, thiamphenicol and sulfamethoxazole.

  1. Line 21, Line 28: Replace ‘’SMA’’ with ‘’SMZ’’ in order to have the same notation in all manuscript body。

 Dear reviewer:

The sorption capacity of GAC towards TC, THI and SMZ were evaluated based on the adsorption kinetics and isotherm. It was found that the pseudo-second-order kinetic model described the sorption of TC,THI, and SMZ on GAC better than the pseudo-first-order kinetic model. According to Langmuir isotherm model, the maximum adsorption capacity of GAC towards TC, THI and SMZ were calculated to be 17.02, 30.40, and 26.77mg/g respectively. Thermodynamic parameters of ΔG°, ΔS°, and ΔH° were obtained, indicating that all the sorptions were spontaneous and exothermic in nature. These results provided a knowledge basement on the utilization of activated carbon to the removal of TC, THI and SMZ from water.

  1. Line 37: Remove the point before ‘’[1]’’Please rewrite the information presented at L54-L56: ‘’ Antibiotics are obtaining the identification of rising environmental contaminants, are classified as fractious bioaccumulative substances[10], as well as are considered as harmful and toxic chemicals.’’ Please add some references for the information: ‘’Its application to the elimination of antibiotics for approximately 30 different compounds have been reported so far.’’ Please check the number of references presented at Line 75: ‘’different kind of activated carbons [20-23, 35].

    Dear reviewer:

According to recent sales data of world market, China is the biggest producer as well as user of antibiotics[1].

Antibiotics are obtaining the identification of rising environmental contaminants, are classified as fractious bio-accumulative substances[10-11], as well as are considered as harmful and toxic chemicals

Different adsorbent materials are used for the elimination of antibiotics from the aqueous solution, for instance, clinoptilolite [19], soil [20], different kind of activated carbons [21-24]

  1. Line 100: Please delete ‘’by using following dilution equation’’
  2. Line 101: Please remove ‘’2)’’ before ‘’Effect of GAC dosage’’
  3. Line 101: Please delete ‘’…and temperature’’
  4. Line 102: Please add that the effect of GAC dosage was also conducted for the sorption of THI and SMZ

    Dear reviewer:

Stock solution preparation: 0.1000 gram of TC, THI, and SMZ were dissolved in a 50-mL volumetric flask with ultra-pure water, and then transfer into a 100-mL volumetric flask to obtain the stock solutions with final concentration of 1g/L respectively. The testing solutions with different concentrations were obtained by diluting the stock solutions with ultrapure water.

Effect of GAC dosage: on the sorption of TC by the GAC, tests were conducted in a 100-mL beaker containing 50mL TC testing solution with 25mg/L. For GAC dosage, the amount of GAC ranged from 2 to 8 g/L, while the temperature was fixed at 25oC and contact time was 100 min,the effect of GAC dosage was also conducted for the sorption of THI and SMZ.

  1. Line 120: Add a point at the end of sentence. The same observation for Line 126
  2. Line 123: Superscript 25oC
  3. Line 127: ‘’Analysis’’ instead of ‘’Aanalysis’’

  1. Line 161-163: Please check the information. Fig. 1(B) shows zeta potential

Dear reviewer:The graph has been corrected.

  1. Line 176: pHZPC instead of Phzpc

  1. Line 208: ‘’and’’ instead of ‘’And’’

  1. Figure 3: Please check: Linear Fit of Sheet1 B"t/qt"; Linear Fit of Sheet1 C"t/qt"; Linear Fit of Sheet1 D"t/qt"; Linear Fit of Sheet1 E"t/qt

  1. Line 246: Fig. 4 instead of Fig. 3

  1. Figure 4. Please correct the x-axis: t1/2 (min1/2) instead of t1/2 (mint1/2)

  1. Figure 5. Please correct the x-axis: Time (min) instead of Time (mint)

  1. Line 281: Table 4 instead of Table 2

  1. Line 291: L/mg instead of litres per milligram.
  2. Line 292: mg/L instead of milligrams per litter
  3. Line 293: mg/g instead of milligrams per gram
  4. Line 294: mg/g instead of milligram per gram

  1. Line 307: Table 5 instead of Table 3

  1. Line 309: ‘’Weber and Chakkravorti’’ instead of ‘’weber and chakkravorti’’

  1. Line 316: Please correct the typo: ‘’Talle (4-5)’’

  1. Line 356: Please check the numbering of the Table

  1. Please add some references from the last 4-5 years.

Dear Reviewer:

The literature for the past five years has been supplemented.

Round 2

Author Response

  1. An example can be found here.

Reviewer 2 Report

My decision is based on the following:

I understand that there are not many papers related to the removal of antibiotics using Granular activated carbon material as the authors mentioned in the new version of the manuscript at Line 79. The authors didn’t specified/highlighted the originality of their research and why their results have an interest for the scientific community compared with the study presented by Fan et al.

Fan et al. studied the capacity of Granular activated carbon to remove Isoniazid, Sulfamethoxazole, Thiamphenicol, and Doxycycline from Aqueous Solution. The authors mentioned in the review note the following:‘’ In this paper, Fan et al. describe in detail the preparation process and characteristics of GAC, and the use of this specific GAC to degrade antibiotics in wastewater is a better application of the GAC prepared by Fan et al.’’

In the present research, the authors investigated the capacity of the material synthesized under the same conditions presented by Fan et al.  to remove Tetracycline, Thiamphenicol and Sulfamethoxazole.

''Dear Reviewer:

The GAC preparation conditions used here are the same as Fan et al.''

So, the adsorption of Sulfamethoxazole and Thiamphenicol are presented in both articles. Consequently, I don’t see the novelty of the present study.

I suggested to add FTIR analysis of GAC sample in the present study. So, I don’t agree with the answer of the authors ‘’ ….we will add FTIR to the method of the next study. Thank you for your suggestion’’ and I consider this section incomplete.

In the first report  I wrote the following: ‘’Please add some references in order to highlight the following information: ‘’A possible reason was that with an increase in GAC dosage, the sorbent surface area, the number of sorption sites, and the contact area increased’’ and unfortunately I didn’t find this observation in the authors report note.

I suggested to add the fit of Pseudo first order kinetic model in the manuscript, but the authors mentioned that: ‘’ I’m so sorry to tell you that the curve fitted by the first-order kinetic model formula is not ideal, so it is not described in the text. I think the question you raised is very correct, but in order to keep the length of the article and the graph within the number of words required by the journal, we give up’’.  I am not agree with this response. In this case, neither Intra-particle diffusion model does not fits the data and the graphs are presented (Figure 4). The data could have been presented in the Supplementary file.

The following two suggestion were not considered. The authors didn’t provide any answer to this comments:

1.                  A comparison between GAC sample and other adsorbents presented in the literature for the removal of TC, THI, and SMZ

2.                  An adsorption/desorption study in order to establish for how many cycles the GAC sample can be utilized for the removal of tetracycline, thiamphenicol and sulfamethoxazole

This observation was not corrected in the new version of the manuscript: ‘’Figure 3: Please check: Linear Fit of Sheet1 B"t/qt"; Linear Fit of Sheet1 C"t/qt"; Linear Fit of Sheet1 D"t/qt"; Linear Fit of Sheet1 E"t/qt’’           What does it represent ‘’Sheet1 B"t/qt"; Sheet1 C"t/qt"; Sheet1 D"t/qt"; Sheet1 E"t/qt’’? The authors must to specify the correct information in the manuscript.

The References section was not modified as I suggested: ‘’Please add some references from the last 4-5 years’’ (observation no 37 from my review report).

Fan et al. is presented as reference 26 at the References Section and not Reference 27 as the authors mentioned in the manuscript at Line 93.

The authors did not give an answer to my comment no 11 from my first review report: ’’The fit of Temkin isotherm is not presented in the case of THI’’.

Author Response

  • I understand that there are not many papers related to the removal of antibiotics using Granular activated carbon material as the authors mentioned in the new version of the manuscript at Line 79. The authors didn’t specified/highlighted the originality of their research and why their results have an interest for the scientific community compared with the study presented by Fan et al.

Fan et al. studied the capacity of Granular activated carbon to remove Isoniazid, Sulfamethoxazole, Thiamphenicol, and Doxycycline from Aqueous Solution. The authors mentioned in the review note the following:‘’ In this paper, Fan et al. describe in detail the preparation process and characteristics of GAC, and the use of this specific GAC to degrade antibiotics in wastewater is a better application of the GAC prepared by Fan et al.’’

In the present research, the authors investigated the capacity of the material synthesized under the same conditions presented by Fan et al.  to remove Tetracycline, Thiamphenicol and Sulfamethoxazole.

''Dear Reviewer:

The GAC preparation conditions used here are the same as Fan et al.''

So, the adsorption of Sulfamethoxazole and Thiamphenicol are presented in both articles. Consequently, I don’t see the novelty of the present study.

Dear Reviewer:

Compared with the study of Fan et al., this paper further discussed the degradation mechanism of GAC to antibiotics at the thermodynamic and kinetic levels, and obtained the optimal temperature, dosage, initial concentration, pH of GAC for the degradation of TC, THI, and SMZ. , and the number of GAC regenerations. Therefore, I think this article is a deeper exploration of the research of Fan et al.

  • I suggested to add FTIR analysis of GAC sample in the present study. So, I don’t agree with the answer of the authors ‘’ ….we will add FTIR to the method of the next study. Thank you for your suggestion’’ and I consider this section incomplete.

Dear Reviewer:

It can be seen from Figure 1(D) that the infrared spectrum of GAC has a C-O characteristic absorption peak near 1000 cm-1, a C=C characteristic absorption peak near 1600 cm-1, and a -OH characteristic absorption peak near 3200 cm-1.

  • In the first report  I wrote the following: ‘’Please add some references in order to highlight the following information: ‘’A possible reason was that with an increase in GAC dosage, the sorbent surface area, the number of sorption sites, and the contact area increased’’ and unfortunately I didn’t find this observation in the authors report note.

Dear Reviewer:

A possible reason was that with an increase in GAC dosage, the sorbent surface area, the number of sorption sites, and the contact area increased.[28-29] 

  1. Zhu, Y. , Liu, L. eds.Experimental study of sewage plant tail water treatment by granular active carbon immobilized catalyst fenton-like. Shandong Chemical Industry 2018.
  2. Qin, Q. , Chen, Y. eds. Optimization of the modified components of mn-sn-ce/gac particle electrode by response surface method. Industrial Water Treatment 2019.
  • I suggested to add the fit of Pseudo first order kinetic model in the manuscript, but the authors mentioned that: ‘’ I’m so sorry to tell you that the curve fitted by the first-order kinetic model formula is not ideal, so it is not described in the text. I think the question you raised is very correct, but in order to keep the length of the article and the graph within the number of words required by the journal, we give up’’.  I am not agree with this response. In this case, neither Intra-particle diffusion model does not fits the data and the graphs are presented (Figure 4). The data could have been presented in the Supplementary file.

Dear Reviewer:

The relevant parameters of the first-order kinetic model have been added to the table below. It can be seen from the table that the R2 of TC is 0.98 when the initial concentration is 24.04 mg/L, and the R2 of THI is 0.76 when the initial concentration is 17.81 mg/L. The R2 of SMZ at an initial concentration of 50.65 mg/L was 0.95.

Pseudo–first-order model

Pseudo–second-order model

Sorbate

C0

(mg/L)

qe,exp

(mg/g)

qe,cal

(mg/g)

K1

(min1)

R2

qe,cal

(mg/g)

K2

[g/(min. mg)]

K2q2 e,cal

[mg/(min. g)]

R2

TC

6.01

0.52

2.332

0.023

0.77

0.49

0.704

0.169

0.68

12.02

1.27

0.481

0.043

0.88

1.64

0.028

0.076

0.91

24.04

2.78

2.773

0.053

0.98

3.46

0.014

0.167

0.98

48.09

5.75

3.377

0.050

0.94

6.27

0.020

0.790

0.99

THI

7.12

0.79

15.7

0.109

0.73

0.80

5.14

3.306

1.00

10.68

1.24

14.11

0.100

0.74

1.25

2.39

3.740

1.00

17.81

2.13

12.23

0.091

0.76

2.14

2.65

12.158

1.00

35.62

4.35

10.51

0.096

0.75

4.37

1.54

29.409

1.00

SMZ

6.33

0.63

4.57

0.021

0.78

0.65

0.811

0.349

0.99

12.66

1.42

3.38

0.027

0.77

1.46

0.482

1.033

0.99

25.32

3

1.11

0.038

0.84

3.10

0.185

1.781

0.99

50.65

6.16

1.58

0.074

0.95

6.4

0.78

31.948

0.99

  • The following two suggestion were not considered. The authors didn’t provide any answer to this comments:
    • A comparison between GAC sample and other adsorbents presented in the literature for the removal of TC, THI, and SMZ

Dear Reviewer:

  • An adsorption/desorption study in order to establish for how many cycles the GAC sample can be utilized for the removal of tetracycline, thiamphenicol and sulfamethoxazole

Dear Reviewer:

Saturated GAC(8.0000±0.0004)g adsorbing TC, THI and SMZ was placed in a quartz glass reactor, and N2 was used as a protective gas and placed in a microwave oven for irradiation, microwave power 730w, microwave time 180s[27], and carried out microwave regeneration test.

Through regeneration experiments, it was found that the adsorption capacity of granular activated carbon for TC, THI, and SMZ decreased with the increase in regeneration times. At an ambient temperature of 25°C, the initial concentration of antibiotics was 25 mg/L, and the dosage of GAC was 8 g/L, the adsorption of TC on GAC reached saturation after 60 minutes. The maximum adsorption efficiencies of initial GAC, 1 GAC regeneration, and 5 regeneration GAC were 92.54%, 85.73%, and 62.14%. Under the same conditions, the maximum adsorption efficiencies of initial GAC, 1-time regeneration GAC, and 5-time regeneration GAC for THI and SMZ were: 96.32%, 32.33%; 91.23%, 85.21%; 70.26%, 50.19%, respectively.

  • This observation was not corrected in the new version of the manuscript: ‘’Figure 3: Please check: Linear Fit of Sheet1 B"t/qt"; Linear Fit of Sheet1 C"t/qt"; Linear Fit of Sheet1 D"t/qt"; Linear Fit of Sheet1 E"t/qt’’           What does it represent ‘’Sheet1 B"t/qt"; Sheet1 C"t/qt"; Sheet1 D"t/qt"; Sheet1 E"t/qt’’? The authors must to specify the correct information in the manuscript.

Dear Reviewer:

The Fig has been modified as follows

  • The References section was not modified as I suggested: ‘’Please add some references from the last 4-5 years’’ (observation no 37 from my review report).

Dear Reviewer:

New references added over the last five years:

  1. Fan, Y.; Zheng, C. eds. Preparation of Granular Activated Carbon and Its Mechanism in the Removal of Isoniazid, Sulfamethoxazole, Thiamphenicol, and Doxycycline from Aqueous Solution. Environ. Eng. Sci. 2019. 36, 1027-1040.
  2. IlavskJ.;BarlokovD.;Marton M. Removal of selected pesticides from water using granular activated carbon. IOP 2021.
  3. Zhu, Y. , Liu, L. eds.Experimental study of sewage plant tail water treatment by granular active carbon immobilized catalyst fenton-like. Shandong Chemical Industry2018.
  4. Qin, Q. , Chen, Y. eds. Optimization of the modified components of mn-sn-ce/gac particle electrode by response surface method. Industrial Water Treatment 2019.

31.Gagliano E.;Falciglia P P.;Zaker Y.et al. Microwave regeneration of granular activated carbon saturated with PFAS. Water research2021,15,198.

  1. 32. Liu, P.; Wang, Q. eds. Sorption of sulfadiazine, norfloxacin, metronidazole, and tetracycline by grangranular activated carbon: Kinetics, mechanisms, and isotherms. Water air and soil pollution journal 2017. 228, 129 1027-1040.
  • Fan et al. is presented as reference 26 at the References Section and not Reference 27 as the authors mentioned in the manuscript at Line 93.
  1. Fan, Y.; Zheng, C. eds. Preparation of Granular Activated Carbon and Its Mechanism in the Removal of Isoniazid, Sulfamethoxazole, Thiamphenicol, and Doxycycline from Aqueous Solution. Environ. Eng. Sci. 2019. 36, 1027-1040.

  • The authors did not give an answer to my comment no 11 from my first review report: ’’The fit of Temkin isotherm is not presented in the case of THI’’.

Dear Reviewer:

The Fig has been modified as follows
